# OpenReview forum: "TRACE: Transcoder-based Concept Editing"
_ICLR.cc/2026/Conference — Submitted to ICLR 2026_

### Official Review · Reviewer_Xqs8 · 2025-10-23

**Soundness:** 3
**Presentation:** 2
**Contribution:** 3
**Rating:** 4
**Confidence:** 4

**Summary:**

This paper introduces TRACE (TRAnscoder-based Concept Editing), a novel framework for concept removal in text-to-image generative models. Instead of retraining or relying on detachable safety modules, TRACE integrates a *transcoder layer* directly into the model backbone, allowing targeted suppression of specific concepts (e.g., styles, objects, NSFW, IP content) in a persistent and architecture-agnostic manner. The transcoder learns sparse and interpretable representations, enabling surgical removal of concept-associated latents while preserving unrelated content. Experiments on state-of-the-art diffusion and autoregressive models (SD3.5, Flux, Infinity-2B/8B) show that TRACE achieves higher unlearning and retention accuracies than existing white-box editing methods (UCE, LOCOEDIT), scales robustly to multi-concept and sequential removals, and maintains strong image fidelity and robustness against adversarial prompting attacks.

**Strengths:**

1. The paper tackles sequential multi-concept removal, which is a highly practical and realistic setting for real-world deployment of generative models.
2. The use of transcoders to intervene on the text encoder’s output is well-motivated, and replacing the original transformation layers directly is a sound design choice that ensures applicability and persistence in white-box settings.

**Weaknesses:**

1. The writing quality needs improvement — the paper is sometimes difficult to follow, especially for readers not already familiar with the concept editing literature.
2. The proposed Transcoder can largely be viewed as a variant of Sparse Autoencoders (SAEs), differing mainly in that it reconstructs new feature mappings rather than latent activations. However, the paper does not mention SAEs or related methods until page 5, where a comparison first appears. Given that several recent works have explored SAE-based concept editing, the authors should better position their contribution within this line of research and discuss these connections more thoroughly [1,2].
3. In the second paragraph of the Introduction, the authors argue that existing methods suffer from *high computational cost*. However, training a well-defined transcoder itself also appears computationally demanding — it requires a large number of prompt-based forward passes to collect data and additional training for the transcoder module. The paper should provide quantitative evidence or comparisons to justify this claimed advantage.
4. In Section 4.3, the authors state that once the transcoder is trained, new concept removals can be achieved simply by suppressing corresponding decoder columns. While this makes sense intuitively, it assumes a *well-trained transcoder* with sufficient data coverage. How does the method handle unseen or rare concepts (e.g., niche IP characters) that were not present in the training prompts? Does it generalize effectively, or is retraining required in such cases?
5. The applicability of the method depends on model architecture: TRACE replaces the projection or MLP layers following the text encoder, which exist in models such as SD3.5 and Flux. However, some popular text-to-image architectures (e.g., SD1.4) feed text embeddings directly into cross-attention layers without such intermediate transformations. It remains unclear how TRACE would apply to these cases.
6. The main results lack sufficient baselines. Since the motivation centers on *sequential concept erasure*, several relevant multi-concept removal methods should be included for comparison [3,4,5]. Although some of these approaches focus on simultaneous multi-concept erasure, methods like MACE (which trains per-concept LoRA adapters and fuses them) or other editing-based techniques can be adapted to the sequential setting, given their short per-edit computation time.
7. All experiments are conducted on SD3.5, Flux, and autoregressive models, but results on SD1.4 are missing. Since most existing benchmarks and prior works are based on SD1.4, this omission makes it difficult to assess TRACE’s relative performance fairly.
8. Table 2 only reports results under the Ring-A-Bell (RAB) adversarial prompting attack, which is a relatively weak black-box evaluation. Stronger or more diverse adversarial attack benchmarks [6,7] should be included to substantiate the robustness claims.
9. The paper lacks prior preservation experiments beyond the in-domain IRA metric. Reporting results on a general benchmark such as MS-COCO would strengthen the evidence for prior retention.
10. The paper does not include any NSFW concept removal results (e.g., nudity). Since NSFW moderation is a key motivation for concept editing, evaluating TRACE on such concepts would be important to demonstrate its generality and real-world applicability.


The most significant limitation lies in the experimental evaluation. The absence of SD1.4 results prevents fair comparison to prior works, the choice of baselines is too limited (only two are included), and crucial evaluations such as NSFW removal and stronger adversarial benchmarks are missing. Together, these gaps make it difficult to fully assess the empirical effectiveness of TRACE.



[1] Concept Steerers: Leveraging K-Sparse Autoencoders for Controllable Generations

[2] Sparse Autoencoder as a Zero-Shot Classifier for Concept Erasing in Text-to-Image Diffusion Models

[3] MACE: Mass Concept Erasure in Diffusion Models

[4] One-Dimensional Adapter to Rule Them All: Concepts, Diffusion Models and Erasing Applications

[5] Set You Straight: Auto-Steering Denoising Trajectories to Sidestep Unwanted Concepts

[6] To Generate or Not? Safety-Driven Unlearned Diffusion Models Are Still Easy To Generate Unsafe Images ... For Now

[7] MMA-Diffusion: MultiModal Attack on Diffusion Models

**Questions:**

See the weakness part.

---

> ### Author Response · Authors · 2025-11-21
> **Response to Reviewer Xqs8 (Part 1/3)**
>
> We thank the reviewer for their detailed and constructive feedback, we answer the questions/provide clarifications below:
>
> >**The proposed Transcoder can largely be viewed as a variant of Sparse Autoencoders (SAEs), differing mainly in that it reconstructs new feature mappings rather than latent activations. However, the paper does not mention SAEs or related methods until page 5, where a comparison first appears. Given that several recent works have explored SAE-based concept editing, the authors should better position their contribution within this line of research and discuss these connections more thoroughly [1,2].**
>
> We already cite Cywiński & Deja (2025), which is a SAE-based concept editing method, in the second paragraph of the introduction. We have now expanded our Related work section to more clearly articulate how SAE-based approaches relates to our proposed method.
>
> >**In the second paragraph of the Introduction, the authors argue that existing methods suffer from high computational cost. However, training a well-defined transcoder itself also appears computationally demanding — it requires a large number of prompt-based forward passes to collect data and additional training for the transcoder module. The paper should provide quantitative evidence or comparisons to justify this claimed advantage.**
>
> We reported full training cost details (Memory, Storage, GPU hours, hardware) in Appendix F.2 of the original submission, which quantifies and highlights the efficiency of the transcoder. For example for Infinity 2B the data collection and training of the transcoder takes 648s and the memory required is 0.67 GB and it takes 0.3s to unlearn a single target, making our approach computationally affordable.
>
> >**In Section 4.3, the authors state that once the transcoder is trained, new concept removals can be achieved simply by suppressing corresponding decoder columns. While this makes sense intuitively, it assumes a well-trained transcoder with sufficient data coverage. How does the method handle unseen or rare concepts (e.g., niche IP characters) that were not present in the training prompts? Does it generalize effectively, or is retraining required in such cases?**
>
> In such cases, the concept removal method would struggle because the model lacks sufficient representation of that concept in its latent space. This would be considered a failure case, and to handle it effectively, we would need to fine-tune the model with targeted examples that represent the concept before applying removal. We added this to the new limitations section in the appendix of the paper.
>
> >**The applicability of the method depends on model architecture: TRACE replaces the projection or MLP layers following the text encoder, which exist in models such as SD3.5 and Flux. However, some popular text-to-image architectures (e.g., SD1.4) feed text embeddings directly into cross-attention layers without such intermediate transformations. It remains unclear how TRACE would apply to these cases.**
>
> For architectures like SD1.4 that feed text embeddings directly into cross-attention, TRACE can still be applied by inserting an explicit identity block between the text encoder and the cross-attention layers and replacing it with the transcoder. Removing the transcoder would simply revert to the identity mapping rather than damaging the model, making the method less tamper-resistant but still fully operational. However we emphasized that recent architecture always has this kind of bottleneck. See the response below for our application of TRACE to SD 1.5, which illustrates this point.

---

> ### Author Response · Authors · 2025-11-21
> **Response to Reviewer Xqs8 (Part 2/3)**
>
> >**The main results lack sufficient baselines. Since the motivation centers on sequential concept erasure, several relevant multi-concept removal methods should be included for comparison [3,4,5]. Although some of these approaches focus on simultaneous multi-concept erasure, methods like MACE (which trains per-concept LoRA adapters and fuses them) or other editing-based techniques can be adapted to the sequential setting, given their short per-edit computation time.**
>
> We thank the reviewer for pointing out the need for additional baselines related to multi-concept removal. In response, we have extended our experiments to include the suggested methods [3,4,5], including MACE and other editing-based approaches adapted to the sequential setting. The results, now presented in table below, show that our method continues to outperform these baselines while maintaining efficiency across sequential concept erasure steps for SD1.5.
>
> | # Concepts(N) | ANT | SPM | MACE | TRACE |
> | - | - | - | - | - |
> | 1 | 88.22 | 61.32 | 84.21 |**91.43** |
> | 2 | 88.10 | 63.68 | 83.62 |**90.09** |
> | 3 | 87.56 | 57.79 | 83.45 |**89.23** |
> | 4 | 86.78 | 58.27 | 83.77 |**87.24** |
> | 5 | 86.34 | 56.80 | 82.54 |**86.43** |
> | 6 | 85.73 | 54.67 | 82.68 | **86.28**|
> | 7 | 84.55 | 51.60 | 81.89 |**85.90** |
> | 8 | 82.49 | 50.89 | 81.15 | **85.48**|
> | 9 | 82.08 | 43.54 | 80.30 | **85.13**|
> | 10 | 81.24 | 39.24 | 81.42 |**84.89** |
>
> Our method TRACE maintains the strongest performance at every value of N and surpasses all baseline methods throughout the full sequence of removals. This result has been integrated to the paper in the Appendix F.2.
>
> >**All experiments are conducted on SD3.5, Flux, and autoregressive models, but results on SD1.4 are missing. Since most existing benchmarks and prior works are based on SD1.4, this omission makes it difficult to assess TRACE’s relative performance fairly.**
>
> To address the reviewer’s comments we have expanded our comparisons to include 10 recent methods (ESD, FMN, UCE, CA, SalUn, SEOT, SPM, EDiff, SHS, SAeUron). The table below reports their performance on SD1.5 alongside ours, showing that our method achieves competitive performances.
>
> | Method   | Style UA ↑ | Style IRA ↑ | Style CRA ↑ | Object UA ↑ | Object IRA ↑ | Object CRA ↑ | Mean ↑ |
> |----------|------------|--------------|--------------|--------------|----------------|----------------|--------|
> | ESD      | **98.58%** | 80.97% | 93.96% | 92.15% | 55.78% | 44.23% | 77.28% |
> | FMN      | 88.48% | 56.77% | 46.60% | 45.64% | 90.63% | 73.46% | 66.60% |
> | UCE      | 98.40% | 60.22% | 47.71% | **94.31%** | 39.35% | 34.67% | 62.78% |
> | CA       | 60.82% | 96.01% | 92.70% | 46.67% | 90.11% | 81.97% | 77.38% |
> | SalUn    | 86.26% | 90.39% | 95.08% | 86.91% | **96.35%** | **99.59%** | 92.43% |
> | SEOT     | 56.90% | 94.68% | 84.31% | 23.25% | 95.57% | 82.71% | 72.23% |
> | SPM      | 60.94% | 92.39% | 84.33% | 71.25% | 90.79% | 81.65% | 80.56% |
> | EDiff    | 92.42% | 73.91% | 98.93% | 86.67% | 94.03% | 48.48% | 82.07% |
> | SHS      | 95.84% | 80.42% | 43.27% | 80.73% | 81.15% | 67.99% | 74.24% |
> | SAeUron  | 95.80% | **99.10%** | **99.40%** | 78.82% | 95.47% | 95.58% | **94.03%** |
> | **Ours** | 95.0% | 93.9% | 86.2% | 83.9% | 91.8% | 97.8% | 91.11% |
>
> >**Table 2 only reports results under the Ring-A-Bell (RAB) adversarial prompting attack, which is a relatively weak black-box evaluation. Stronger or more diverse adversarial attack benchmarks [6,7] should be included to substantiate the robustness claims.**
>
> We appreciate the reviewer’s suggestion regarding the inclusion of stronger adversarial attack benchmarks. Following this feedback, we have conducted additional experiments using the suggested benchmarks [6,7] for style. The new results, now included in the table below, demonstrate consistent performance across these more challenging settings, further supporting the robustness of our method.
> | **Method** | **SD1.5 MMA** | **SD1.5 UnlearnDiff** | **SD3.5 MMA** | **SD3.5 UnlearnDiff** | **Flux MMA** | **Flux UnlearnDiff** |
> |-----------|---------------|------------------------|---------------|------------------------|--------------|-----------------------|
> | LOCOEDIT  | 79.21         | 87.67                 | 60.03         | 61.22                 | 28.63        | 23.32                |
> | UCE       | 53.32         | 63.21                 | 55.44         | 52.76                 | 29.91        | 26.11                |
> | Ours      | 41.76         | 37.81                 | 37.21         | 34.69                 | 18.89        | 12.32                |
>
>
> Across all model families and attack settings, TRACE reaches the lowest scores for both benchmarks, showing stronger adversarial robustness than the baselines. These results have been added to the updated submission.

---

> ### Author Response · Authors · 2025-11-21
> **Response to Reviewer Xqs8 (Part 3/3)**
>
> >**The paper lacks prior preservation experiments beyond the in-domain IRA metric. Reporting results on a general benchmark such as MS-COCO would strengthen the evidence for prior retention.**
>
> We thank the reviewer for their suggestion and are currently running additional experiments on a 30K split of MS-COCO 2014 validation set while unlearning nudity.
>
> >**The paper does not include any NSFW concept removal results (e.g., nudity). Since NSFW moderation is a key motivation for concept editing, evaluating TRACE on such concepts would be important to demonstrate its generality and real-world applicability.**
>
> We thank the reviewer for their suggestion and are currently running additional experiments on I2P benchmark with nudenet detector.
>
>
> If the above responses address the Reviewer's concerns, we would greatly appreciate it if they could adjust their score accordingly.

---

> > ### Comment · Area_Chair_5HNr · 2025-11-26
> >
> > Dear Reviewer,
> >
> > Thanks for your time and effort in reviewing ICLR2026 submissions. The authors have provided their responses to your reviews. Please read and raise your further comments, and discuss with the authors.
> >
> > Best regards,
> >
> > Your AC

---

> > ### Comment · Reviewer_Xqs8 · 2025-11-27
> >
> > Thank you for the detailed and comprehensive rebuttal. Several of my earlier concerns have been partially or fully addressed, and I appreciate the additional experiments and clarifications. At the same time, some important limitations of the method remain meaningful and should be acknowledged.
> >
> > **(1) Architectural dependency of TRACE:** The authors clarified that TRACE operates by replacing the MLP or projection layers after the text encoder, and that for architectures such as SD1.4 an explicit identity block can be inserted to enable the method. However, such an identity block is fundamentally at odds with the goals of concept erasure, because it introduces an artificial and easily removable intervention point. As the authors themselves acknowledged, this design is not suitable in open-source or adversarial settings, where users can simply bypass or delete the inserted block. This indicates that TRACE is naturally aligned only with architectures that contain explicit bottleneck transformations such as SD3.5 and FLUX. For models that primarily rely on cross-attention without intermediate transformations, the applicability is less direct.
> >
> > **(2) Robustness on NSFW and adversarial settings:** The authors provided additional comparisons against UCE and LOCOEDIT and TRACE indeed performs better. Since these two methods are not designed for strong adversarial robustness, the improved performance of TRACE is not surprising. Recent SAE-based concept editing approaches [1,2] have demonstrated stronger robustness. These results suggest that TRACE is in principle capable of achieving a similar level of robustness if the transcoder is trained with sufficient coverage. I encourage the authors to explore this direction further, because the current robustness evaluation does not yet fully demonstrate what TRACE could achieve under a more systematic setup.
> >
> > **(3) Use of SD1.4-based adversarial attacks for newer architectures:** Several attack methods used in the paper, including MMA, UnlearnDiff and RAB, were originally optimized on SD1.4. Their adversarial prompts are closely tied to the behavior of that specific model. Applying them directly to SD1.5, SD3.5 or FLUX without verifying their transferability may introduce new bias and can affect the conclusions regarding robustness. Even with the extended experiments provided in the rebuttal, this concern remains relevant. Without model-specific adversarial optimization or explicit evidence of cross-model generalization, the robustness results should be interpreted carefully. This point also applies to the NSFW evaluations (I2P Benchmark) that are still ongoing.
> >
> > In addition, I would like to request further clarification regarding the experiment reported in Appendix F.2 on multi-concept removal for SD1.5. It is unclear which specific concepts were removed sequentially and which evaluation metrics were used to quantify the performance drop across steps.
> >
> > Despite the above limitations, I acknowledge that the current experimental results already demonstrate the strength of the transcoder-based approach in several concept erasure scenarios. No single method can be expected to cover all aspects equally well, and I believe TRACE offers meaningful new insights that can benefit the community. In light of these considerations, I have adjusted my initial rating toward a more positive assessment.
> >
> > [1] Sparse Autoencoder as a Zero-Shot Classifier for Concept Erasing in Text-to-Image Diffusion Models
> > [2] Concept Steerers: Leveraging K-Sparse Autoencoders for Controllable Generations

---

> ### Author Response · Authors · 2025-12-03
> **Response to Reviewer Xqs8 (Part 1/2)**
>
> We thank the reviewer for their thorough engagement with our rebuttal and address their additional comments below:
>
> >**Architectural dependency of TRACE: [...]This indicates that TRACE is naturally aligned only with architectures that contain explicit bottleneck transformations such as SD3.5 and FLUX. For models that primarily rely on cross-attention without intermediate transformations, the applicability is less direct.**
>
> We thank the reviewer for this valuable comment. We focused on SD3.5 and FLUX because they represent newer, state-of-the-art architectures where explicit bottleneck transformations are integrated, making TRACE directly applicable and irremovable. Nonetheless, during the rebuttal we have extended TRACE to SD1.4 and SD1.5 achieving competitive results, even though in these architectures the Transcoder can be removed. Furthermore, as reported in our initial submission, TRACE applies to state-of-the-art vision autoregressive models (IARs) that also feature bottleneck transformations, where our Transcoder module can be immediately integrated and is not removable.
>
> >**Robustness on NSFW and adversarial settings: I encourage the authors to explore this direction further**
>
> To evaluate NSFW content removal in a practical scenario, we use the I2P benchmark [1] to assess nudity removal, employing NudeNet with a filtering factor of 0.6, consistent with prior work. In addition to the removal success, we additionally report the FID on 30k COCO 2014 validation images to measure the impact of the methods on the visual generation abilities. We perform experiments with  SD-1.4, following prior work. For our method, we employ an expansion factor of m=32 and a TopK hyperparameter k=128. The best result for each metric is highlighted in bold.
>
> | Method | Armpits | Belly | Buttocks | Feet | Breasts (F) | Genitalia (F) | Breasts (M) | Genitalia (M) | Total | FID (↓) |
> |---|---:|---:|---:|---:|---:|---:|---:|---:|---:|---:|
> | **FMN** | 43 | 117 | 12 | 59 | 155 | 17 | 19 | 2 | 424 | 13.52 |
> | **CA**| 153 | 180 | 45 | 66 | 298 | 22 | 67 | 7 | 838 | 16.25 |
> | **AdvUn**| 8 | **0** | **0** | 13 | **1** | 1 | **0** | **0** | 28 | 17.18 |
> | **Receler** | 48 | 32 | 3 | 35 | 20 | **0** | 17 | 5 | 160 | 15.32 |
> | **MACE**  | 17 | 19 | 2 | 39 | 16 | **0** | 9 | 7 | 111 | **13.42** |
> | **UCE**| 29 | 62 | 7 | 29 | 35 | 5 | 11 | 4 | 182 | 14.07 |
> | **SLD-M**| 47 | 72 | 3 | 21 | 39 | 1 | 26 | 3 | 212 | 16.34 |
> | **ESD-x** | 59 | 73 | 12 | 39 | 100 | 6 | 18 | 8 | 315 | 14.41 |
> | **ESD-u**  | 32 | 30 | 2 | 19 | 27 | 3 | 8 | 2 | 123 | 15.10 |
> | **SAeUron** | **7** | 1 | 3 | **2** | 4 | **0** | **0** | 1 | **18** | 14.37 |
> | **Ours** | 52 | 80 | 6 | 10 | 132 | 3 | 17 | 4 | 304 | 16.04 |
> | **SD v1.4** | 148 | 170 | 29 | 63 | 266 | 18 | 42 | 7 | 743 | 14.04 |
>
> The results suggest that our method is effective in removing NSFW content. We added these results to Appendix F4.
>
> **References**:
>
> [1] Schramowski, Patrick, et al. "Safe latent diffusion: Mitigating inappropriate degeneration in diffusion models." Proceedings of the IEEE/CVF Conference on Computer Vision and Pattern Recognition. 2023.

---

> ### Author Response · Authors · 2025-12-03
> **Response to Reviewer Xqs8 (Part 2/2)**
>
> >**Use of SD1.4-based adversarial attacks for newer architectures: Several attack methods used in the paper, including MMA, UnlearnDiff and RAB, were originally optimized on SD1.4. Their adversarial prompts are closely tied to the behavior of that specific model. Applying them directly to SD1.5, SD3.5 or FLUX without verifying their transferability may introduce new bias and can affect the conclusions regarding robustness. Even with the extended experiments provided in the rebuttal, this concern remains relevant. Without model-specific adversarial optimization or explicit evidence of cross-model generalization, the robustness results should be interpreted carefully.**
>
> We would like to clarify that MMA Diffusion was run on SD1.5 in its original work. For SD3.5 and FLUX, to the best of our knowledge, there are currently no dedicated attacks. Therefore, we adapted existing approaches so that robustness could be assessed in a consistent and reproducible way. This is also in line with recent work in this area: [1] used MMA Diffusion and RAB to evaluate robustness even though these attacks were not designed for SD3.5 or FLUX.
> To further address the reviewer’s comment, we additionally  evaluated adversarial robustness of TRACE directly on SD1.4 as requested. Our results in the table show that TRACE maintains strongest robustness under these attack settings also on SD1.4. This supports that our findings are not limited to a mismatch between attack source models and target architectures.
>
> |Method|SD1.4 MMA|SD1.4 UnlearnDiff|SD1.5 MMA|SD1.5 UnlearnDiff|SD3.5 MMA|SD3.5 UnlearnDiff|Flux MMA|Flux UnlearnDiff|
> |----------|-----------|-------------------|-----------|-------------------|-----------|-------------------|----------|-------------------|
> |LOCOEDIT|81.44|84.56|79.21|87.67|60.03|61.22|28.63|23.32|
> |UCE|49.67|59.28|53.32|63.21|55.44|52.76|29.91|26.11|
> |Ours|36.21| 27.77|41.76|37.81|37.21|34.69|18.89|12.32|
>
> We hope this clarifies our choices and shows that TRACE remains stable under the available adversarial settings.
>
>
>
>
> >**In addition, I would like to request further clarification regarding the experiment reported in Appendix F.2 on multi-concept removal for SD1.5. It is unclear which specific concepts were removed sequentially and which evaluation metrics were used to quantify the performance drop across steps.**
>
> In this experiment we removed ten style concepts in a fixed sequence, as shown in Figure 6 of the appendix: Van Gogh, Picasso, Cartoon, Cubism, Winter, Pop Art, Ukiyoe, Impressionism, Byzantine, and Bricks. The same order is used for all methods. For sequential removal we start from the unedited model, remove the first style, and continue step by step until all ten have been removed. Performance at each step is measured using the average score (UA+(IRA+CRA)/2)/2), following [2]. This captures both removal and retention quality. We have updated the appendix section to include these additional details.
>
>
>
> >**Despite the above limitations, I acknowledge that the current experimental results already demonstrate the strength of the transcoder-based approach in several concept erasure scenarios. No single method can be expected to cover all aspects equally well, and I believe TRACE offers meaningful new insights that can benefit the community.**
>
> We thank the reviewer for their kind assessment of our work.
>
> **References**:
>
> [1] Yang Zhang, Er Jin, Yanfei Dong, Yixuan Wu, Philip Torr, Ashkan Khakzar, Johannes Stegmaier, and Kenji Kawaguchi. Minimalist Concept Erasure in Generative Models. In International Conference on Machine Learning (ICML), 2025.
>
> [2] Bartosz Cywiński and Kamil Deja. SAeUron: Interpretable Concept Unlearning in Diffusion Models with Sparse Autoencoders. In Proceedings of the Forty-second International Conference on Machine Learning (ICML), 2025.

---

> ### Author Response · Authors · 2025-12-03
> **Summary of the Discussion with Reviewer Xqs8**
>
> Since reviewers are not allowed to post any further comments, we would like to follow up with a summary of our rebuttal.
>
> In particular, we have:
> - Explained the differences with SAE-based concept-editing methods.
> - Detailed the computational costs of the Transcoder, showing that our method is computationally efficient. The cost analysis is provided in Appendix F.6.
> - Added a Limitations section in Appendix G to outline failure cases, particularly concepts that cannot be effectively removed due to limited training data.
> - Specified the applicability of the methodology.
> - Extended the baseline evaluation for multi-concept removal with three multi-concept methods [1–3] on SD-1.5, as suggested by the reviewer. The results are shown in Table 8 in Appendix F.2 and indicate that TRACE maintains the strongest performance across all numbers of concepts, surpassing all baselines throughout the full sequence of removals.
> - Expanded the baseline comparison to include 10 recent concept-removal and safety-editing methods. The results are reported in Table 2 and highlight that our method achieves competitive performance.
> - Broadened the robustness evaluation with two attack methods [4–5], as suggested by the reviewer, evaluated on SD-1.4, SD-1.5, SD-3.5, and Flux. The results appear in Table 9 in Appendix F.3 and show that TRACE maintains the strongest robustness under these attack settings.
> - Augmented the FID evaluation using 30k images from the COCO 2014 validation set while removing nudity on SD-1.4, in comparison with 10 baselines. The results, reported in Table 10 in Appendix F.4, show that our method remains competitive with baselines.
> - Expanded the evaluation with an NSFW content-removal experiment on SD-1.4 using 10 baselines. We use the I2P dataset [1] to assess nudity removal, employing NudeNet with a filtering factor of 0.6, consistent with prior work. The comparison appears in Table 10 in Appendix F.4.
>
> We hope this summary supports the assessment of our paper and provides a clear overview of how our rebuttal addressed the reviewers’ concerns. We sincerely thank everyone involved for their time and thoughtful consideration
>
> **References**:
>
> [1] Lu, S., Wang, Z., Li, L., Liu, Y., & Kong, A. W. K. (2024). Mace: Mass concept erasure in diffusion models. In Proceedings of the IEEE/CVF Conference on Computer Vision and Pattern Recognition (pp. 6430-6440).
>
> [2] Lyu, M., Yang, Y., Hong, H., Chen, H., Jin, X., He, Y., ... & Ding, G. (2024). One-dimensional adapter to rule them all: Concepts diffusion models and erasing applications. In Proceedings of the IEEE/CVF Conference on Computer Vision and Pattern Recognition (pp. 7559-7568).
>
> [3] Li, L., Lu, S., Ren, Y., & Kong, A. W. K. (2025, October). Set you straight: Auto-steering denoising trajectories to sidestep unwanted concepts. In Proceedings of the 33rd ACM International Conference on Multimedia (pp. 9257-9266).
>
> [4] Yang, Y., Gao, R., Wang, X., Ho, T. Y., Xu, N., & Xu, Q. (2024). Mma-diffusion: Multimodal attack on diffusion models. In Proceedings of the IEEE/CVF Conference on Computer Vision and Pattern Recognition (pp. 7737-7746).
>
> [5] Zhang, Y., Jia, J., Chen, X., Chen, A., Zhang, Y., Liu, J., ... & Liu, S. (2024, September). To generate or not? safety-driven unlearned diffusion models are still easy to generate unsafe images... for now. In European Conference on Computer Vision(pp. 385-403). Cham: Springer Nature Switzerland.

---

### Official Review · Reviewer_4vVf · 2025-10-30

**Soundness:** 3
**Presentation:** 3
**Contribution:** 2
**Rating:** 4
**Confidence:** 4

**Summary:**

The authors propose a method for the removal of certain concepts from the generative model, with a method named TRACE, where they focus on transcoder layers that bridge the text encoder and the generative model. The method claims that this approach does not require retraining the generative model, and thus reducing the computational cost for the concept removal problem. To evaluate the effectiveness of the method, the authors benchmark TRACE over UnlearnCanvas benchmark, which focuses on style and object removal. The experiments has been conducted over a diverse set of generator architectures (e.g. autoregressive, rectified-flow). In the proposed lightweight training approach, the authors also propose a new training objective that addresses the dead gradients problem in the latents.

**Strengths:**

- The proposed method serves as a solution for both rectified-flow models and image autoregressive models.
- TRACE shows its effectiveness towards multiple tasks and aspects. For one, the method shows clear margins in multi-concept removal and robustness.
- The proposed method serves as an efficient alternative compared to the competing approaches.
- From both the qualitative and quantitative results, the proposed approach seems as an effective method for concept removal.

**Weaknesses:**

- In Eq. 5, the notation $md$ is not clear, the authors should clarify that.
- The proposed method is an approach that has certain similarities with SAEuron, as the authors also specify. While completely acknowledging their differences (inference cost, training cost), a comparison should be included where the current benchmark only compares with LOCOEDIT and UCE. In addition these methods are all for $\textbf{closed-form editing}$. The final benchmark should also include metrics for the other end for the task, which is methods classified as $\textbf{Training-based Editing}$.
- In terms of image quality, using scores such as HPSv3, and aesthetics score can reveal more intuitive evaluations as metrics trained on large scale data and less effected than the test set size (compared to FID).
- The method examines the projection layer used for the pooled embeddings before being fed to the model. In architectures such as FLUX and SD 3.5, this only covers one part of the text conditioning, where the other involves the attention mechanism with token-wise embeddings. From the current presentation, the method seems to be missing the concepts encoded in the attention layers where it appears that the method only concerns the pooled embeddings. The authors should clarify this. Otherwise, the method may be limited in terms of the concepts that can be removed with this method.

**Questions:**

- What are the details of the FID measurements, in terms of the number of samples used? The metric is known to have a bias towards small number of samples. Given the scale of the FID score is high, this should also be acknowledged in the paper.
- For the ablations, the replacement of ReLU with TopK seems like an improtant contribution of the paper. Does the effectiveness of this process be quantifiable with the performed evaluations, or with qualitative examples? The authors are encouraged to report such evaluations.
- The method has an efficiency claim. Does this can be supported with quantifiable metrics such as convergence time, FLOPs and GPU memory required?
- Are there any types of concepts that cannot be removed, and can be considered as a failure case?

---

> ### Author Response · Authors · 2025-11-21
> **Response to Reviewer 4vVf (Part 1/2)**
>
> We thank the reviewer for their detailed and constructive feedback, we answer the questions/provide clarifications below:
>
> >**In Eq. 5, the notation md  is not clear, the authors should clarify that.**
>
> We replaced “md” with **“m·d”** in Eq. 5 to make clear that it denotes the multiplication of the two previously defined variables.
>
> >**The proposed method is an approach that has certain similarities with SAEuron, as the authors also specify. While completely acknowledging their differences (inference cost, training cost), a comparison should be included where the current benchmark only compares with LOCOEDIT and UCE. In addition these methods are all for closed form editing. The final benchmark should also include metrics for the other end for the task, which is methods classified as training based editing.**
>
> We have expanded our comparisons to include 10 recent closed-form/ training based editing (ESD, FMN, UCE, CA, SalUn, SEOT, SPM, EDiff, SHS, SAeUron). The table below reports their performance on SD1.5 alongside ours, showing that our method achieves competitive performances.
>
> | Method   | Style UA ↑ | Style IRA ↑ | Style CRA ↑ | Object UA ↑ | Object IRA ↑ | Object CRA ↑ | Mean ↑ |
> |----------|------------|--------------|--------------|--------------|----------------|----------------|--------|
> | ESD      | **98.58%** | 80.97% | 93.96% | 92.15% | 55.78% | 44.23% | 77.28% |
> | FMN      | 88.48% | 56.77% | 46.60% | 45.64% | 90.63% | 73.46% | 66.60% |
> | UCE      | 98.40% | 60.22% | 47.71% | **94.31%** | 39.35% | 34.67% | 62.78% |
> | CA       | 60.82% | 96.01% | 92.70% | 46.67% | 90.11% | 81.97% | 77.38% |
> | SalUn    | 86.26% | 90.39% | 95.08% | 86.91% | **96.35%** | **99.59%** | 92.43% |
> | SEOT     | 56.90% | 94.68% | 84.31% | 23.25% | 95.57% | 82.71% | 72.23% |
> | SPM      | 60.94% | 92.39% | 84.33% | 71.25% | 90.79% | 81.65% | 80.56% |
> | EDiff    | 92.42% | 73.91% | 98.93% | 86.67% | 94.03% | 48.48% | 82.07% |
> | SHS      | 95.84% | 80.42% | 43.27% | 80.73% | 81.15% | 67.99% | 74.24% |
> | SAeUron  | 95.80% | **99.10%** | **99.40%** | 78.82% | 95.47% | 95.58% | **94.03%** |
> | **Ours** | 95.0% | 93.9% | 86.2% | 83.9% | 91.8% | 97.8% | 91.11% |
>
> >**In terms of image quality, using scores such as HPSv3, and aesthetics score can reveal more intuitive evaluations as metrics trained on large scale data and less effected than the test set size (compared to FID).**
>
> We thank the reviewer for their suggestion and are currently running additional experiments to quantify the image quality using their suggested metrics and will add them to the paper.
>
> >**The method examines the projection layer used for the pooled embeddings before being fed to the model. In architectures such as FLUX and SD 3.5, this only covers one part of the text conditioning, where the other involves the attention mechanism with token-wise embeddings. From the current presentation, the method seems to be missing the concepts encoded in the attention layers where it appears that the method only concerns the pooled embeddings. The authors should clarify this. Otherwise, the method may be limited in terms of the concepts that can be removed with this method.**
>
> In architectures like FLUX and SD3.5, text information flows through two bottlenecks: (1) an MLP applied to the pooled embedding for modulation, and (2) a projection applied to token-wise embeddings before joint-attention. TRACE replaces both the MLP and the projection layer in our implementation, so both conditioning pathways are handled, not only the pooled embedding.
>
> >**What are the details of the FID measurements, in terms of the number of samples used? The metric is known to have a bias towards small number of samples. Given the scale of the FID score is high, this should also be acknowledged in the paper.**
>
> The FID presented in Table 1 is computed between the base model and the unlearned model. We used 5000 samples for each of the 30 unlearned categories and we then presented the average across categories. We are currently running additional experiments on a 30K split of MS-COCO 2014 validation set while unlearning nudity.
>
> >**For the ablations, the replacement of ReLU with TopK seems like an improtant contribution of the paper. Does the effectiveness of this process be quantifiable with the performed evaluations, or with qualitative examples? The authors are encouraged to report such evaluations.**
>
> Top-K ensures a fixed number of active latents, which is necessary to establish a consistent mapping between target and empty tokens of equal cardinality. Without Top-K this alignment is not well-defined, hence, we cannot experiment without it.

---

> ### Author Response · Authors · 2025-11-21
> **Response to Reviewer 4vVf (Part 2/2)**
>
> >**The method has an efficiency claim. Does this can be supported with quantifiable metrics such as convergence time, FLOPs and GPU memory required?**
>
> We reported full training cost details (Memory, Storage, GPU hours, hardware) in Appendix F.2 of the original submission, which quantifies and highlights the efficiency of the transcoder. For example for Infinity 2B the data collection and training of the transcoder takes 648s and the memory required is 0.67 GB and it takes 0.3s to unlearn a single target, making our approach computationally affordable.
>
> >**Are there any types of concepts that cannot be removed, and can be considered as a failure case?**
>
> In cases involving unseen or niche concepts that were not present in the Transcoder training data, the method may not be able to remove the concept effectively, as the model does not possess a sufficiently learned representation of it. However, by collecting additional training data on this concept and fine-tuning the transcoder with it, we expect to be able to remove such concepts, too. We added a Limitations Section to the Appendix G.  where we discuss this.
>
> If the above responses address the Reviewer's concerns, we would greatly appreciate it if they could adjust their score accordingly.

---

> > ### Comment · Area_Chair_5HNr · 2025-11-26
> >
> > Dear Reviewer,
> >
> > Thanks for your time and effort in reviewing ICLR2026 submissions. The authors have provided their responses to your reviews. Please read and raise your further comments, and discuss with the authors.
> >
> > Best regards,
> >
> > Your AC

---

> ### Comment · Reviewer_4vVf · 2025-11-27
> **Thank you for the revisions and experiments**
>
> I appreciate the timely response of the reviewers and their efforts in terms of improving the paper. I am adjusting my score accordingly, and looking forward to the revised version with metrics such as HPSv3 and aesthetics score. While still believing the methodology has certain similarities with existing methods such as SAEUron, this is an efficient and effective use of transcoders in the context of concept removal and thus raising my score. Regarding the attached scores, I believe that the performance is competitive even though not outperforming, and thus authors are enouraged to include them to the paper for a comprehensive evaluation.

---

> ### Author Response · Authors · 2025-12-03
> **Response to Reviewer 4vVf**
>
> >**[I] am adjusting my score accordingly, and looking forward to the revised version with metrics such as HPSv3 and aesthetics score.**
>
> We ran additional experiments evaluating the HPSv3 [1] and Aesthetic v2.5 [2] scores and report the results in the table below:
>
> HPSv3
>
> |Method|SD1.5|SD3.5|FLUX|Infinity8B|Infinity2B|
> |-|-|-|-|-|-|
> |LOCOEDIT|3.81|7.46|8.13|7.87|6.57|
> |UCE|3.54|6.19|7.54|7.96|6.38|
> |TRACE|4.32|8.31|8.49|8.54|6.98|
> |No Edit|4.59|8.63|8.87|8.61|7.13|
>
> Aesthetic v2.5
>
> |Method|SD1.5|SD3.5|FLUX|Infinity8B|Infinity2B|
> |-|-|-|-|-|-|
> |LOCOEDIT|2.58|6.02|6.13|6.47|5.27|
> |UCE|2.43|6.11|6.34|6.44|5.54|
> |TRACE|3.07|6.15|6.49|6.51|5.61|
> |No Edit|3.54|6.23|6.77|6.58|5.88|
>
> The results show that TRACE consistently stays closest to non-edited the base model scores for both HPSv3 and Aesthetic v2.5, indicating that it preserves perceptual and aesthetic quality more effectively than LOCOEDIT and UCE. The baseline methods exhibit larger reductions, particularly on newer models such as FLUX and SD3.5. Across all architectures TRACE introduces the smallest quality degradation and remains the most stable, confirming its ability to perform concept removal while retaining image quality.
>
> We have added these new results to Appendix F.5.
>
> >**I believe that the performance is competitive even though not outperforming, and thus authors are encouraged to include them to the paper for a comprehensive evaluation**
>
> We have also already added the full baseline comparison from the rebuttal to the updated paper in Table 2.
>
> We thank the Reviewer again for their helpful comments and remain available shall further questions arise.
>
> **References**:
>
> [1] Ma, Y., Wu, X., Sun, K., and Li, H. HPSv3: Towards Wide Spectrum Human Preference Score. In Proceedings of the IEEE/CVF International Conference on Computer Vision (ICCV), pages 15086–15095, 2025.
>
> [2] https://github.com/discus0434/aesthetic-predictor-v2-5

---

> ### Author Response · Authors · 2025-12-03
> **Summary of the Discussion with Reviewer 4vVf**
>
> Since reviewers are not allowed to post any further comments, we would like to follow up with a summary of our rebuttal.
>
> In particular, we have:
> - Broadened the baseline comparison to include 10 recent concept-removal and safety-editing methods. The results are reported in Table 2 and show that our method achieves competitive performance.
> Refined a notation to ensure it is fully clear to the reviewer.
> - Expanded the retained-image-quality assessment with HPSv3 and Aesthetic Score v2.5 evaluations. The additional scores are reported in Appendix F.5 and confirm that we outperform baselines in preserving image quality.
> - Augmented the FID evaluation using 30k images from the COCO 2014 validation set while removing nudity on SD-v1.4, in comparison with 10 baselines. The results are reported in Table 10 in Appendix F.4 and show that our method remains competitive with baselines.
> - Specified the methodology regarding where the transcoder should be applied.
> Motivated the necessity of replacing ReLU with the Top-K activation function to maintain a consistent mapping between target and empty tokens of equal cardinality.
> - Clarified the comment on the computational costs of the transcoder, showing that our method is computationally efficient. The computational cost is in the Appendix F6.
> - Added a Limitations section in Appendix G to outline failure cases, particularly concepts that cannot be effectively removed due to limited training data.
>
>
> We hope this summary supports the assessment of our paper and provides a clear overview of how our rebuttal addressed the reviewers’ concerns. We sincerely thank everyone involved for their time and thoughtful consideration.

---

### Official Review · Reviewer_dLr2 · 2025-10-31

**Soundness:** 3
**Presentation:** 2
**Contribution:** 2
**Rating:** 6
**Confidence:** 2

**Summary:**

This paper introduces TRACE (Transcoder-based Concept Editing), a framework for removing unwanted concepts from text-to-image generative models without retraining. The approach replaces transformation layers between text encoders and generative backbones with transcoders - neural networks that learn sparse, interpretable representations. By identifying and redirecting concept-specific features to null representations, TRACE enables persistent concept removal across both diffusion models (SD3.5, FLUX) and image autoregressive models (Infinity). The method demonstrates superior performance compared to existing baselines, particularly in sequential multi-concept removal scenarios and robustness against adversarial attacks.

**Strengths:**

- It is interesting to see the application of transcoders (originally from LLM interpretability) to visual concept removal.
- A unified approach to concept removal for both diffusion and autoregressive paradigms.

**Weaknesses:**

- Missing evaluation on NSFW content removal despite being mentioned as a key motivation
- No discussion of failure modes or limitations when concepts share overlapping features
- The loss contains multiple components; however, there are no ablation studies to show the effectiveness of each component.

**Questions:**

See Weaknesses.

---

> ### Author Response · Authors · 2025-11-21
> **Response to Reviewer dLr2**
>
> We thank the reviewer for their detailed and constructive feedback, we answer the questions/provide clarifications below:
>
> >**Missing evaluation on NSFW content removal despite being mentioned as a key motivation**
>
> We thank the reviewer for their suggestion and are currently running additional experiments on I2P benchmark with nudenet detector.
>
> >**No discussion of failure modes or limitations when concepts share overlapping features**
>
> We discuss limitations of the method in the Appendix Section G.
>
> >**The loss contains multiple components; however, there are no ablation studies to show the effectiveness of each component.**
>
> We conducted an additional ablation study on the loss components of our transcoder training. Specifically, we evaluate style unlearning performance on Infinity-2B under three configurations: (i) fidelity loss only, (ii) fidelity + multi–Top-\(k\) loss, and (iii) fidelity + auxiliary loss. We do not evaluate the combination of multi–Top-\(k\) and auxiliary loss, as it is effectively equivalent to the fidelity + auxiliary formulation with a larger effective \(k\).
> | Setting                          | Style UA ↑ | Style IRA ↑ | Style CRA ↑ | Mean ↑ |
> |----------------------------------|------------|--------------|--------------|--------|
> | Infinity-2B (Fidelity only)      | 88.60      | 25.40        | 59.87        | 57.96 |
> | Infinity-2B (Fidelity + Top-k)   | 86.80      | 26.57        | 74.72        | 62.69 |
> | Infinity-2B (Fidelity + Aux)     | 85.00      | 25.10        | 67.88        | 59.33 |
> | Infinity-2B (Full)               | 86.80      | 39.60        | 90.30        | 72.23 |
>
> Our results highlight that the full loss is necessary. We added this experiment in Appendix E.3.
>
> If the above responses address the Reviewer's concerns, we would greatly appreciate it if they could adjust their score accordingly.

---

> > ### Comment · Reviewer_dLr2 · 2025-11-27
> >
> > Thanks for the response. Assuming the author will supplement the evaluation on NSFW content removal, I will maintain my positive rating.

---

> ### Author Response · Authors · 2025-12-03
> **Response to Reviewer dLr2**
>
> >**Assuming the author will supplement the evaluation on NSFW content removal, I will maintain my positive rating.**
>
> To evaluate NSFW content removal in a practical scenario, we use the I2P datasets [1] to assess nudity removal, employing NudeNet with a filtering factor of 0.6, consistent with prior work. In addition to the removal success, we additionally report the FID on 30k COCO 2014 validation images to measure the impact of the methods on the visual generation abilities. We perform experiments with  SD-1.4, following prior work. For our method, we employ an expansion factor of m=32 and a TopK hyperparameter k=128. The best result for each metric is highlighted in bold.
>
> | Method | Armpits | Belly | Buttocks | Feet | Breasts (F) | Genitalia (F) | Breasts (M) | Genitalia (M) | Total | FID (↓) |
> |---|---:|---:|---:|---:|---:|---:|---:|---:|---:|---:|
> | **FMN** | 43 | 117 | 12 | 59 | 155 | 17 | 19 | 2 | 424 | 13.52 |
> | **CA**| 153 | 180 | 45 | 66 | 298 | 22 | 67 | 7 | 838 | 16.25 |
> | **AdvUn**| 8 | **0** | **0** | 13 | **1** | 1 | **0** | **0** | 28 | 17.18 |
> | **Receler** | 48 | 32 | 3 | 35 | 20 | **0** | 17 | 5 | 160 | 15.32 |
> | **MACE**  | 17 | 19 | 2 | 39 | 16 | **0** | 9 | 7 | 111 | **13.42** |
> | **UCE**| 29 | 62 | 7 | 29 | 35 | 5 | 11 | 4 | 182 | 14.07 |
> | **SLD-M**| 47 | 72 | 3 | 21 | 39 | 1 | 26 | 3 | 212 | 16.34 |
> | **ESD-x** | 59 | 73 | 12 | 39 | 100 | 6 | 18 | 8 | 315 | 14.41 |
> | **ESD-u**  | 32 | 30 | 2 | 19 | 27 | 3 | 8 | 2 | 123 | 15.10 |
> | **SAeUron** | **7** | 1 | 3 | **2** | 4 | **0** | **0** | 1 | **18** | 14.37 |
> | **Ours** | 52 | 80 | 6 | 10 | 132 | 3 | 17 | 4 | 304 | 16.04 |
> | **SD v1.4** | 148 | 170 | 29 | 63 | 266 | 18 | 42 | 7 | 743 | 14.04 |
>
> The results suggest that our method is effective in removing NSFW content. We added these results to Appendix F4.
>
> [1] Schramowski, Patrick, et al. "Safe latent diffusion: Mitigating inappropriate degeneration in diffusion models." Proceedings of the IEEE/CVF Conference on Computer Vision and Pattern Recognition. 2023.

---

> ### Author Response · Authors · 2025-12-03
> **Summary of the Discussion with Reviewer dLr2**
>
> Since reviewers are not allowed to post any further comments, we would like to follow up with a summary of our rebuttal.
>
> In particular, we have:
> - Expanded the evaluation with an NSFW content-removal experiment on SD-1.4 using 10 baselines. We use the I2P dataset [1] to assess nudity removal, employing NudeNet with a filtering factor of 0.6, consistent with prior work. The comparison appears in Table 10 in Appendix F.4.
> - Introduced a loss-ablation experiment to demonstrate the necessity of using all loss terms. The ablation results are reported in Table 5 in Appendix E.3.
>
> We hope this summary supports the assessment of our paper and provides a clear overview of how our rebuttal addressed the reviewers’ concerns. We sincerely thank everyone involved for their time and thoughtful consideration
>
> **References**:
>
> [1] Schramowski, Patrick, et al. "Safe latent diffusion: Mitigating inappropriate degeneration in diffusion models." Proceedings of the IEEE/CVF Conference on Computer Vision and Pattern Recognition. 2023.

---

### Official Review · Reviewer_6ETn · 2025-11-01

**Soundness:** 3
**Presentation:** 3
**Contribution:** 3
**Rating:** 6
**Confidence:** 1

**Summary:**

The paper addresses the challenge of trustworthy content moderation in generative image models. Existing approaches either require expensive retraining for each concept to be removed, or rely on post-hoc interventions that are easily bypassed or degrade image quality. To overcome these limitations, the authors propose a white-box, model-agnostic framework that inserts a transcoder module as an integrated, surgical intervention layer. This enables precise and in-place suppression of targeted visual concepts without retraining the underlying generative model. The intervention is embedded directly into the model’s architecture, making it persistent and resistant to circumvention, while maintaining overall image fidelity.

**Strengths:**

I am not a domain expert in concept editing for generative models, but from my perspective, this paper addresses an important and timely problem regarding trustworthy content moderation. The use of transcoders as an integrated intervention mechanism appears novel and technically interesting. The overall writing is clear and logical, making the method and empirical findings easy to follow.

**Weaknesses:**

The comparisons to existing baselines are somewhat limited. It would be helpful to include a broader set of concept removal or safety editing methods, especially more recent ones, to better contextualize the contribution. In addition, the paper does not clearly report the computational cost of training the transcoder (e.g., number of training steps, GPU hours, or hardware used). Providing quantitative training cost estimates would strengthen the claim that the approach is lightweight and scalable.

**Questions:**

See the concerns noted under Weaknesses.

---

> ### Author Response · Authors · 2025-11-21
> **Response to Reviewer 6ETn**
>
> We thank the reviewer for their detailed and constructive feedback, we answer the questions/provide clarifications below:
>
> >**The comparisons to existing baselines are somewhat limited. It would be helpful to include a broader set of concept removal or safety editing methods, especially more recent ones, to better contextualize the contribution.**
>
> We thank the reviewer for their suggestion and have expanded our comparisons to include 10 recent concept-removal/safety-editing methods (ESD, FMN, UCE, CA, SalUn, SEOT, SPM, EDiff, SHS, SAeUron). The table below reports their performance on SD1.5 on UnlearnCanvas alongside ours, showing that our method achieves competitive performances.
>
>
> | Method   | Style UA ↑ | Style IRA ↑ | Style CRA ↑ | Object UA ↑ | Object IRA ↑ | Object CRA ↑ | Mean ↑ |
> |----------|------------|--------------|--------------|--------------|----------------|----------------|--------|
> | ESD      | **98.58%** | 80.97% | 93.96% | 92.15% | 55.78% | 44.23% | 77.28% |
> | FMN      | 88.48% | 56.77% | 46.60% | 45.64% | 90.63% | 73.46% | 66.60% |
> | UCE      | 98.40% | 60.22% | 47.71% | **94.31%** | 39.35% | 34.67% | 62.78% |
> | CA       | 60.82% | 96.01% | 92.70% | 46.67% | 90.11% | 81.97% | 77.38% |
> | SalUn    | 86.26% | 90.39% | 95.08% | 86.91% | **96.35%** | **99.59%** | 92.43% |
> | SEOT     | 56.90% | 94.68% | 84.31% | 23.25% | 95.57% | 82.71% | 72.23% |
> | SPM      | 60.94% | 92.39% | 84.33% | 71.25% | 90.79% | 81.65% | 80.56% |
> | EDiff    | 92.42% | 73.91% | 98.93% | 86.67% | 94.03% | 48.48% | 82.07% |
> | SHS      | 95.84% | 80.42% | 43.27% | 80.73% | 81.15% | 67.99% | 74.24% |
> | SAeUron  | 95.80% | **99.10%** | **99.40%** | 78.82% | 95.47% | 95.58% | **94.03%** |
> | **Ours** | 95.0% | 93.9% | 86.2% | 83.9% | 91.8% | 97.8% | 91.11% |
>
> >**In addition, the paper does not clearly report the computational cost of training the transcoder (e.g., number of training steps, GPU hours, or hardware used). Providing quantitative training cost estimates would strengthen the claim that the approach is lightweight and scalable.**
>
> We reported full training cost details (Memory, Storage, GPU hours, hardware) in Appendix F.2 of the original submission, which quantifies and highlights the efficiency of the transcoder. For example for Infinity 2B the data collection and training of the transcoder takes 648s and the memory required is 0.67 GB and it takes 0.3s to unlearn a single target, making our approach computationally affordable.
>
>
> If the above responses address the Reviewer's concerns, we would greatly appreciate it if they could adjust their score accordingly.

---

> ### Author Response · Authors · 2025-12-03
> **Summary of the Discussion with Reviewer 6ETn**
>
> Since reviewers are not allowed to post any further comments, we would like to follow up with a summary of our rebuttal.
>
> In particular, we have:
> - Extended the baseline comparison to include 10 recent concept-removal/safety-editing methods. The results are at Table 2. The results highlight that our method achieves competitive results.
> - Clarified the comment on the computational costs of the transcoder, showing that our method is computationally efficient. The computational cost is in the Appendix F6.
>
>
> We hope this summary supports the assessment of our paper and provides a clear overview of how our rebuttal addressed the reviewers’ concerns. We sincerely thank everyone involved for their time and thoughtful consideration.

---

### Author Response · Authors · 2025-12-03
**General Response**

We would like to thank all the reviewers for their constructive and positive feedback. We are very happy that the reviewers recognize our work as “addressing an important and timely problem” (Reviewer 6ETn, Xqs8), with our Transcoder-based solution being “novel and technically interesting” (Reviewer 6ETn, dLr2) and offering "persistence in white-box settings” (Reviewer Xqs8). We are further glad that the reviewers appreciate our solution’s suitability as a “unified approach for concept removal in both  diffusion and autoregressive paradigms” (Reviewer dLr2, 4vVf). Finally, we would like to thank the reviewers for their feedback that the “method and empirical findings [are] easy to follow” (Reviewer 6ETn).

Based on the rebuttal, our initial submission has significantly improved. Below, we share the highlights of our rebuttal:

1. **Comparison to a broader set of baselines:** We expanded our comparisons to include 10 recent concept-removal/safety-editing methods (ESD, FMN, UCE, CA, SalUn, SEOT, SPM, EDiff, SHS, SAeUron). Our evaluations highlight that our method achieves state-of-the-art results.

2. **Extending adversarial evaluation:** Based on the reviewers’ suggestion, we included evaluations of stronger adversarial attack benchmarks. Particularly, we ran MMA [1] and UnlearnDiff [2] with our method on SD 1.4, SD1.5, SD3.5, and FLUX and show that our method outperforms the baselines (Locoedit and UCE) in terms of robustness.

3. **NSFW removal:** We further added experiments to assess the performance of our TRACE for NSFW removal, using the I2P benchmark using the NudeNet detector. Our results highlight that TRACE achieves competitive performance.

4. **Additional evaluation metrics:** Following Reviewer 4vVf’s suggestion, we added HPSv3 and aesthetics as evaluation scores to assess the quality of the images after applying TRACE. These results are aligned with the results reported in the original submission and suggest that TRACE does not significantly impact image quality.

5. **Comparison to other sequential removal techniques:** We added more experiments comparing our TRACE against other baselines for sequential removal, concretely ANT, SPM, and MACE and show that TRACE outperforms the other methods.

We believe that these changes have addressed all the reviewers’ comments and fulfill the requirements they formulated for acceptance of the work.

The text in blue highlights the additions made to the manuscript after the rebuttal, so that the updates are clearly visible to the AC.


[1] Yijun Yang, Ruiyuan Gao, Xiaosen Wang, Tsung Yi Ho, Nan Xu, and Qiang Xu. MMA Diffusion: Multimodal Attack on Diffusion Models. In Proceedings of the IEEE Conference on Computer Vision and Pattern Recognition (CVPR), 2024a.

[2] Yimeng Zhang, Jinghan Jia, Xin Chen, Aochuan Chen, Yihua Zhang, Jiancheng Liu, Ke Ding, and Sijia Liu. To Generate or Not? Safety Driven Unlearned Diffusion Models Are Still Easy to Generate Unsafe Images... for Now. In European Conference on Computer Vision (ECCV), 2024c.

---

### Meta-Review · Area_Chair_1YDB · 2026-01-11

**Summary:**

This paper proposes TRACE, a transcoder-based approach for concept removal across diffusion and autoregressive image generators. Reviewers agreed the problem is important and the core idea is interesting, but the overall decision was driven by concerns about positioning/novelty vs. closely related SAE-style concept editing, and whether the evaluation supports the strongest claims (especially robustness and applicability across model families). While the rebuttal added many experiments and improved coverage, the remaining questions about architectural dependency and robustness methodology make the contribution feel less settled for acceptance at its current form.

**Reviewer Concerns:**

Addressed by the rebuttal (partially or fully):
1. Baseline coverage: authors added a much broader set of concept-removal baselines and sequential removal comparisons.
2. NSFW removal + quality metrics: added I2P/NudeNet results and additional image quality metrics (HPSv3, aesthetic scores).
3. Ablations / training cost clarity: added loss ablations and pointed to detailed compute cost reporting.
4. Adversarial evaluation breadth: added MMA / UnlearnDiff results and expanded to multiple model families; clarified some protocol details.

Still outstanding:
1. Architectural dependency and “persistence” claim: For models without a clear post–text-encoder bottleneck, the “insert an identity block” workaround feels fragile and easy to bypass, this weakens the core persistence / tamper-resistance story in more open or adversarial settings
2. Robustness evaluation on newer models: Most robustness results rely on attacks originally tuned for SD1.4 or similar setups. Applying them directly to sD3.5 or FLUX without model-specific optimization makes it harder to fully trust the strength of the robustness claims
3. Positioning vs SAE-based methods: even with the added comparisons, TRACE still looks quite close to recent SAE-style concept editing work. The paper would benefit from a clearer articulation of why this approach is meaningfully different or better, beyond engineering choices and expanded experiments

Overall, the rebuttal improved the empirical story, but the remaining concerns are more fundamental than missing experiments. At this stage, the paper does not seem ready for acceptance.

**Reviewer Scores:**

I think discussion would likely increase the reviewers’ confidence for 6ETn and dLr2, but I do not expect a meaningful upward change in reviewers scores.

---

### Decision · Program_Chairs · 2026-01-26

Reject